# Effects of Dietary Xylanase and Arabinofuranosidase Combination on the Growth Performance, Lipid Peroxidation, Blood Constituents, and Immune Response of Broilers Fed Low-Energy Diets

**DOI:** 10.3390/ani9070467

**Published:** 2019-07-22

**Authors:** Ahmed A. Saleh, Abeer A. Kirrella, Safaa E. Abdo, Mahmoud M. Mousa, Nemat A. Badwi, Tarek A. Ebeid, Ahmed L. Nada, Mahmoud A. Mohamed

**Affiliations:** 1Department of Poultry Production, Faculty of Agriculture, Kafrelsheikh University, Kafr El-Sheikh 333516, Egypt; 2Department of Animal Wealth Development, Faculty of Veterinary Medicine, Kafrelsheikh University, Kafr El-Sheikh 333516, Egypt; 3Department of Animal Production and Breeding, College of Agriculture and Veterinary Medicine, Qassim University, Buraydah 51452, Saudi Arabia; 4Orkila Egypt Chemicals SAE, 12 Al-Badeya Street, El Merghany Heliopolis, Cairo 11757, Egypt; 5Adisseo France SAS 10 Place du Général de Gaulle, Antony, 92160 Paris, France

**Keywords:** xylanase, arabinofuranosidases, broilers, nutrient utilization, growth performance, immunity

## Abstract

**Simple Summary:**

Arabinoxylans (AXs) constitute the major non-starch polysaccharides (NSPs) existent in maize and soybean meal, comprising about 52% and 65% of the total NSP. Previous works have illustrated that the incorporation of arabinofuranosidase (Abf; GH51) plus xylanase (Xyl; GH11) enhanced the dry matter digestibility of maize and wheat in vitro, in comparison with Xyl alone. In broilers, the combination of dietary Xyl and Abf (Rovabio^®^ Advance) enhanced energy, fat, fiber, and protein utilizations. This study shows the effect of feeding low-energy diets with or without Rovabio^®^ Advance, including high concentrations of Xyl and Abf, on the growth performance, nutrient digestibility, lipid peroxidation, blood constituents, and immune response of broilers. Our results confirm the improved growth, digestibility, and immunity obtained by enzymes supplementation. Furthermore, diets supplemented with enzymes caused a higher antibody titer against the Newcastle disease virus. Moreover, they enhanced plasma lipid profiles and antioxidation.

**Abstract:**

The present study was conducted to examine that impact of dietary xylanase (Xyl) and arabinofuranosidase (Abf) supplementation on the performance, protein and fat digestibility, the lipid peroxidation, the plasma biochemical traits, and the immune response of broilers. A total of 480, un-sexed, and one-day-old broilers (Ross 308) were randomly divided into three treatments with eight replicates, where chicks in the first treatment were fed basal diets and served as the control, chicks in the second treatment were fed diets formulated with reductions of 90 kcal/kg, and chicks in the third treatment were fed the same formulated diets used in the second group as well as the Xyl and Abf combination (Rovabio^®^ Advance). Feed intake was decreased by the low energy diet, leading to an enhancement in feed efficiency enzyme supplementation in the low energy diet (*p* < 0.015). Both protein and fat digestibility were improved (*p* < 0.047) due to enzyme supplementation. Moreover, enzyme supplementation increased muscle total lipids content and decreased muscle thiobarbituric acid retroactive substance content. Furthermore, diets supplemented with Xyl and Abf exhibited an increase in antibody titers against the Newcastle disease virus (*p* < 0.026). In addition, enzyme supplementation increased gene expression related to growth and gene expression related to fatty acid synthesis. It could be concluded that dietary Xyl and Abf supplementation had beneficial impacts on growth, nutrient digestibility, lipid peroxidation, immune response, and gene expressions related to growth and fatty acid synthesis in broiler chickens fed low-energy diets.

## 1. Introduction

Nowadays, in the broiler production industry, the total price of energy ingredients is about 65–70% of the total costs of the broiler diets. Additionally, these ingredients are usually imported from outside Egypt. Therefore, diverse experiments have been conducted to decrease the cost by reducing the rate of these energy ingredients in broiler feed, along with animating the performance of broilers [1]. One scenario involves adding enzymes to the broiler diets, which promotes such growth performance parameters as feed efficiency and body weight gain [2]. Indeed, the presence of soluble non-starch polysaccharides (NSPs) has reduced nutrient utilization and consequently minimized growth performance in broilers. These carbohydrates cannot be digested by birds, as they do not have the capability to produce these enzymes. Therefore, NSP enzymes are functional when they are added to cereal-based diets, e.g., wheat, soybean, barley, and maize. Exogenous enzymes like xylanase (Xyl), amylase, and protease are produced using a microbial source [3]. Almirall et al. found that the feed conversion ratio (FCR) was enhanced by exogenous enzyme supplementation in broiler diets, and this effect was connected with improving the digestibility and minimizing the viscosity of intestinal contents [4]. Furthermore, Abdel-Latif et al. and Saleh et al. illustrated that the improvement of growth performance due to NSP enzyme addition might be explained by their participation in reducing the digesta viscosity and amendment of gut microbiota by improving the beneficial microbes [3,5]. Worldwide broiler production, including in Egypt, farmers depend on corn and soybean meal for feeding birds, as these are the available ingredients; however, the level of NSP is 29% in soybean and 9% in corn [6]. Broilers do not produce enzymes for the hydrolysis of these NSPs present in the cell walls of the grains [7,8]. Exogenous enzymes which can hydrolyze NSPs are abundantly found in the feed given to birds [9]. 

Arabinoxylans (AXs) exemplify the major NSP existent in maize and wheat, comprising about 4.7% and 7.3% of the dry matter and 65% and 52% of the total NSP, respectively [10]. Previous studies showed that a mixture of arabinofuranosidase (Abf) and Xyl improved the dry matter digestibility of maize and wheat in vitro, in comparison with Xyl alone [11]. In fact, endoxylanases support the degradation of AXs by hydrolyzing the xylan backbone. Additionally, arabinose substitutions minimized the activity of Xyl in yellow corn and its correlating byproducts [10]. Abf could split the xylose backbone in arabinose and give access to endoxylanase activity [12]. Moreover, Cozannet et al. illustrated that dietary Rovabio^®^ Advance (including high concentrations of Xyl and Abf) had a positive effect on the energy utilization and digestibility of starch, protein, fat, and insoluble and soluble fibers [11]. Furthermore, Ravn et al. documented that the addition of an enzyme combination (Xyl and Abf) improved the growth performance and gut morphology in broilers [13]. It could be hypothesized that the supplementation of a combination of Xyl and Abf to broiler diets might be involved in improving the utilization of nutrients of a low-energy diet and could consequently enhance the growth performance and immune responsiveness of broilers. The objective of this study was to examine the impact of dietary Xyl and Abf (Rovabio^®^ Advance) supplementation on the growth performance, fat and protein digestibility, lipid peroxidation, biochemical plasma traits, and immune response of broilers fed low-energy diets.

## 2. Materials and Methods

### 2.1. Animals and Experimental Design

The study was approved by the Ethics Committee of Local Experimental Animals Care Committee and conducted in accordance with the guidelines of Kaferelsheikh University, Egypt (Number 4/2016 EC). A total of 480 un-sexed one-day-old broilers (Ross 308) were housed in bins (stocking density was 10 birds/m^2^) and randomly divided into 3 experimental treatments with 8 replicates (20 birds each) to equalize the average body weight in each group. The control group was fed basal diets as commercial feed formulated according to the strain requirements, the second experimental group of chicks was fed diets formulated with reductions of 90 kcal/kg AME and 3% digestible amino acids, and the third experimental group was fed the same formulated diets used in the second group as well as Xyl and Abf. The composition and chemical analysis of the experimental diets (starter, grower, and finisher) are shown in Table 1. The Xyl and Abf (Rovabio^®^ Advance) were kindly given by the Adisseo company, France S.A.S. Antony Parc 210, Place du Général de GaulleF-92160 ANTONY, France. ® This enzyme was industrially created by the fermentation of *Talaromyces versatilis* (IMI378536 and DSM26702; Adisseo France S.A.S. proprietary strains), and the main enzyme activity in Enz comes from Xyl and Abf. The enzyme was added to the premix mixture, which is one of the basic ingredients in the all diets. The diets were provided to the birds ad libitum; starter diets were in crumble form. However, the grower and finisher diets were pellet form. The trail was managed in an open-door house with a 23 h light–1 h dark cycle. Daily temperature and humidity inside the house were controlled at 24–26 °C and 60–70%, respectively. The experimental diets were offered from 1 day to 35 days of age. Bird body weight was measured individually every week. However, feed intake was measured daily (on a group basis per pen) throughout the experimental period. At 32 days, all birds were weighed individually and sorted from the smallest to the heavy weight. Then, 12 male birds/treatment have the same average weight were transferred to special batteries containing individual cages to enact the digestibility experiment. Then, these birds were slaughtered and dissected to gauge the weights of the breast muscle, thigh muscle, liver, gizzard, heart, spleen, abdominal fat, and bursa of Fabricius. All organs were weighed and described as a ratio of the body weight. Blood samples were collected from the wing vein immediately before slaughtering, gathered into heparinized test tubes, and then rapidly centrifuged (3000 rpm for 20 min at 5 °C) to separate the plasma. Plasma was stored at −20 °C pending analysis.

### 2.2. Nutrient Digestibility

In the last three days of the experiment, excreta were gathered and weighted from 12 males per treatment, where broilers were housed individually in special metabolic cages (40 × 40 × 50 cm) for digestibility tests. During these three days, the birds and feed intake were weighted daily, and extracted faces were collected, weighted, and stored in a freezer. After the digestibility experiment period, all samples were dried in a drying oven at 60 °C for 24 h. The whole dried samples were then homogenized. Samples were taken and finely ground for analysis according to the Association of Official Analytical Chemists (AOAC) [14]. The crude protein concentration in the diet and excreta was gauged to determine nitrogen digestibility using the Kjeldahl method, and crude fat was gauged by the Soxhlet method (AOAC 945.38 F and 920.39 C, respectively). The calculation was as follows: Nitrogen digestibility (%) = (total nitrogen intake − total nitrogen excreted)/total nitrogen intake × 100.

### 2.3. Biochemical Analysis

Triglycerides (TG), total cholesterol, high-density lipoprotein (HDL) cholesterol, low-density lipoprotein (LDL) cholesterol, glutamic oxaloacetic transaminase (GOT), glutamate pyruvate transaminase (GPT), glucose, creatinine, total protein, albumin, and globulin were measured colorimetrically using commercial kits (Diamond Diagnostics, Egypt) according to the procedure outlined by the manufacturer. Muscle total lipid content, fatty acid profile, and amino acid analysis were measured using gas liquid chromatography (GLC) according to the method of Saleh [15]. The muscle thiobarbituric acid retroactive substance (TBARS) concentration was measured by the process of Ohkawa et al. [16]. 

### 2.4. Serum Antibody Titers

Serum antibody titers against Newcastle disease (ND) and avian influenza (H9N1) were determined by means of the hemagglutination inhibition test using standard methods qualified by OIE [17].

### 2.5. RNA Analysis

Each breast muscle sample was homogenized, and the total RNA was extracted using a total RNA purification kit following the manufacturer’s protocol (Fermentas, K0731, Thermo Fisher Scientific, Waltham, MA, USA). The extracted total RNA (5 µg per sample) was reverse transcribed into cDNA using Revert Aid H Minus Reverse Transcriptase as described by the manufacturer (Fermentas, EP0451, Thermo Fisher Scientific, Waltham, MA, USA). Following amplification, PCR products were electrophoresed, and the expression level of different bands was analyzed using the ImageJ gel analysis program [18]. 

### 2.6. Statistical Analysis

The differences between the experimental treatments and the control were analyzed with a General Liner model using SPSS Statistics 17.0 (Statistical Packages for the Social Sciences, SPSS Inc., Chicago, IL, USA, released 23 August 2008). Tukey’s multiple comparison test was used to identify which treatment conditions were significantly different from each other.

## 3. Results and Discussion

One of the major objectives of the current study was to evaluate the impacts of feeding low-energy diets supplemented with or without Xyl and Abf enzymes on the growth performance, nutrient digestibility, lipid peroxidation, blood plasma biochemical traits, immune response, and gene expressions related to growth and fatty acid synthesis in broilers. The inclusion of Xyl and Abf enzymes in low-energy diets in the present study improved the growth performance in broilers, and this improvement might be related to the enhancement of nutrient digestibility by Xyl and Abf enzyme supplementation. This supposition is in harmony with Nortey et al., who suggested that dietary exogenous enzyme addition had a beneficial effect on nutrient digestibility in swine specimens [19]. Moreover, Slominski et al. reported that the inclusion of a debranching enzymes mixture improved the overall enzyme effectiveness and, consequently, enhanced the nutrient digestibility and the alleviation of the negative impacts of NSPs [20]. Recently, Ravn et al. stated that the addition of an enzyme combination (Xyl and Abf) improved duodenum villi length, which was probably involved in enhancing the growth performance, including the body weight and FCR, in broilers [13].

The data presented in Table 2 show that feeding low-energy diets decreased body weight gain significantly compared to the control group, while, dietary supplementation with Xyl and Abf enzymes increased body weight gain and improved FCR, crude protein digestibility, and crude fat digestibility significantly (*p* < 0.05). No significant differences were detected in the feed intake. These findings are in correspondence with Cozannet et al., who demonstrated that a dietary combination of Xyl and Abf had a positive effect on the energy utilization and digestibility of protein, starch, fat, and insoluble and soluble fibers [11]. Additionally, Cowieson and Ravindran reported improvements in crude protein and amino acid digestibility when a multiple enzyme mixture including protease, Xyl, and amylase was employed to supplement corn–soybean diets [21]. Similarly, Rutherfurd et al. found enhancements in crude protein and amino acid digestibility in broilers fed commercial diets supplemented with a multiple enzymes complex, including amylase, β-glucanase, and Xyl [22]. Cowieson and Ravindran reported that the mechanisms that enhanced the amino acid utilization due to the addition of exogenous enzymes are connected with minimizing endogenous losses related to a decreased secretion of endogenous enzymes [23]. Moreover, Meng et al. stated that dietary enzymes take off the nutrient encapsulating effect of NSP [24], thus enhancing the nutrient availability to endogenous enzymes and improving the overall nutrient digestibility and intestinal microbial environment [11,21].

The bursa of Fabricius relative weight was significantly increased by feeding low-energy diets supplemented with Xyl and Abf enzymes, while the abdominal fat relative weight was significantly decreased in both low-energy diets and control. However, the carcass, thigh, liver, gizzard, heart, and spleen relative weights were not affected by low-energy diets supplemented with or without a combination of Xyl and Abf enzymes (Table 3). These findings are in agreement with previous reports [25,26]. However, Farran et al. found that breast muscle, pectoralis major, thigh, and drum yields were not affected by the inclusion of enzyme preparations [27]. Garipoglu et al. reported that the dressing percentage was reduced but abdominal fat weight was not influenced by feeding diets supplemented with multienzymes [28]. Similarly, Kocher et al. found that found that a Xyl, amylase and protease addition to the corn–soybean meal did not affect abdominal fat weight [29]. Contrarily, Garcia et al. reported that the Xyl and β-glucanase supplementation of barley–wheat-based diets elevated the abdominal fat content in broilers [30]. In the present study (Table 3), the lower abdominal fat relative weight noted in low-energy diets, with or without enzyme supplementation, might be attributed to the fact that the lower energy diets caused less fattening and were also connected with reducing the feed intake.

Table 4 shows the effect of low-energy diets supplemented with or without Xyl and Abf enzymes on blood biochemical parameters. Plasma globulin and HDL-cholesterol were significantly increased due to dietary supplementation in comparison with the control group. However, plasma total cholesterol was significantly reduced by the addition Xyl and Abf enzymes to low-energy diets compared with the control group; however, plasma GOT, GPT, albumin, triglycerides, glucose, LDL-cholesterol, and creatinine were not significantly affected. Interestingly, serum antibody titers against ND were significantly increased by the enzyme group, while the antibody titer against avian influenza (H9N1) was enhanced insignificantly in comparison with the control group (*p* = 0.11). However, there was an insignificant increase in the antibody titer against avian influenza (H9N1) due to the dietary Xyl and Abf enzyme supplementation to low-energy diets. These results supported our findings of the bursa of Fabricius relative weight, which was significantly increased due to the dietary Xyl and Abf enzyme supplementation. These results are coincident with our previous findings, which noted that ND and infectious bronchitis disease (IBD) antibody titers were improved by enzyme addition to broiler diets, and this may be regarded as an improvement in protein digestibility because of these enzyme mixtures and peptide transporter 1(PEPT1) gene expression, which enhanced absorption [3]. Different authors have reported the impacts of supplementing AX or arabinoxylooligosaccharides in a broiler diet, and they have observed that oligosaccharides with a polymerization score of less than five encourage the propagation of beneficial bacteria and enhance microbiota variety [31,32,33]. The increment of arabinoxylooligo-saccharides (AXOS) presented in wheat or soybean AX improves the proliferation of bifidobacteria in ceca without influencing the body weight of birds. The inclusion of AXOS encourages beneficial bacteria and protects against pathogenic bacteria [34,35], which consequently enhances gut health. Lei et al. documented that dietary Xyl, Abf, and feruloyl esterase improved gut health [36]. Pettey et al. also indicated that adding 0.05% β-mannanase and arabinoxylooligosaccharides led to an improved blood IGF-I concentration in growing and finishing pigs [37]. It might be speculated that the inclusion of Xyl and Abf in low-energy diets had a positive effect on gut health and nutrient digestibility, leading to improved growth performance, lymphoid organs weights, and immune response in broilers.

Liver function indicators (plasma GOT and GPT) and the kidney function indicator (plasma creatinine) were not significantly affected. These results are in harmony with Ahmad et al., who evaluated the effect of dietary Xyl addition on plasma biochemical constituents in broilers and illustrated that Xyl might be safe in poultry rations without negative effects on vital organ functions [38]. Additionally, Saleh et al. reported that the serum concentrations of GOT, GPT, and creatinine were not significantly affected by dietary enzyme supplementation [3].

The data presented in Table 5 illustrate the effect of a low-energy diet supplemented with or without Xyl and Abf enzymes on the muscle content of fatty acids, amino acids, total lipids, and TBARS. Muscle TBARS content was significantly decreased, while muscle total lipids content was significantly increased by feeding a low-energy diet supplemented with Xyl and Abf enzymes compared with the control group. However, the muscle contents of lysine, methionine, oleic, linoleic, and linolenic acids were not significantly influenced by dietary treatments. These findings are in agreement with Cowieson and Ravindran, who found enhancements in the digestibility of lysine, methionine, cysteine, and threonine when a multiple enzyme mixture possessing protease, Xyl, and amylase was used to supplement corn-based diets, but these improvements did not affect the amino acid contents in muscle [21]. Furthermore, Head et al. reported that dietary α-linolenic acids in the form of linseed resulted in a significant increase of hepatic omega-3 poly unsaturated fatty acids (n-3PUFA) [39]; however, the inclusion of a multiple enzyme complex of Xyl and amylase in a linseed-based diet resulted in a reduction in the n-6PUFA-like linoleic acid, but oleic and linolenic acids were not affected. Regarding lipid peroxidation, the data illustrate that the muscle TBARS concentration was decreased by reducing the energy in diets, and this agreed with Cho and Kim, who observed that muscle malondialdehyde (MDA) concentration was decreased in low-energy density diets supplemented with or without β-mannanase and Xyl supplementation in pigs [9]. 

The mRNA expressions of the growth hormone receptor (GHR), insulin-like growth factor receptor (IGFR), and fatty acid synthesis (FAS) were significantly increased by adding Xyl and Abf enzymes to low-energy diets in comparison with the control group (Figure 1A–C). Meanwhile, acetyl-coA carboxylase (ACC) was increased by Xyl and Abf enzyme supplementation to a low energy diet but these differences were not significant in comparison with the control group (Figure 1D). These improvements are confirmed by the previous results of Guo et al., who demonstrated that the addition of Xyl upregulated the expression of the sodium–glucose cotransporter 1 and IGFR genes in broiler chickens [40]. Furthermore, Hosseini et al. reported that the inclusion of Xyl improved the expression of GHR and IGFR genes [41]. Moreover, ACC and FAS encode a biotin-dependent enzyme which is involved in the biosynthesis of fatty acids via the catalyzation of the irreversible carboxylation of acetyl-CoA for malonyl-CoA production [42]. In the present study, ACC and FAS gene expressions were significantly elevated in the low-energy diet supplemented with the Xyl and Abf enzyme combination. This impact was connected with the freedom of blocked macronutrients and, consequently, elevated lipogenesis due to the eternal energy adequacy [43]. Indeed, FAS and ACC, which play important roles in the lipogenic passage, are also key determinants for the maximal ability of a muscle tissue to synthesize fatty acid; they are extremely expressed in tissues such as those of the liver and muscles [3,44]. In common physiological cases, nutritional factors such as high-fat feed and hormones could organize the enzyme activity and gene expression of the FAS and ACC [39,45,46].

## 4. Conclusions

It could be concluded that dietary Xyl and Abf (Rovabio^®^ Advance) supplementation had positive effects on the growth performance, protein and fat digestibility, plasma lipid profiles, lipid peroxidation, immune response, and gene expression related to growth and fatty acid synthesis in broiler chickens fed low-energy diets.

## Figures and Tables

**Figure 1 animals-09-00467-f001:**
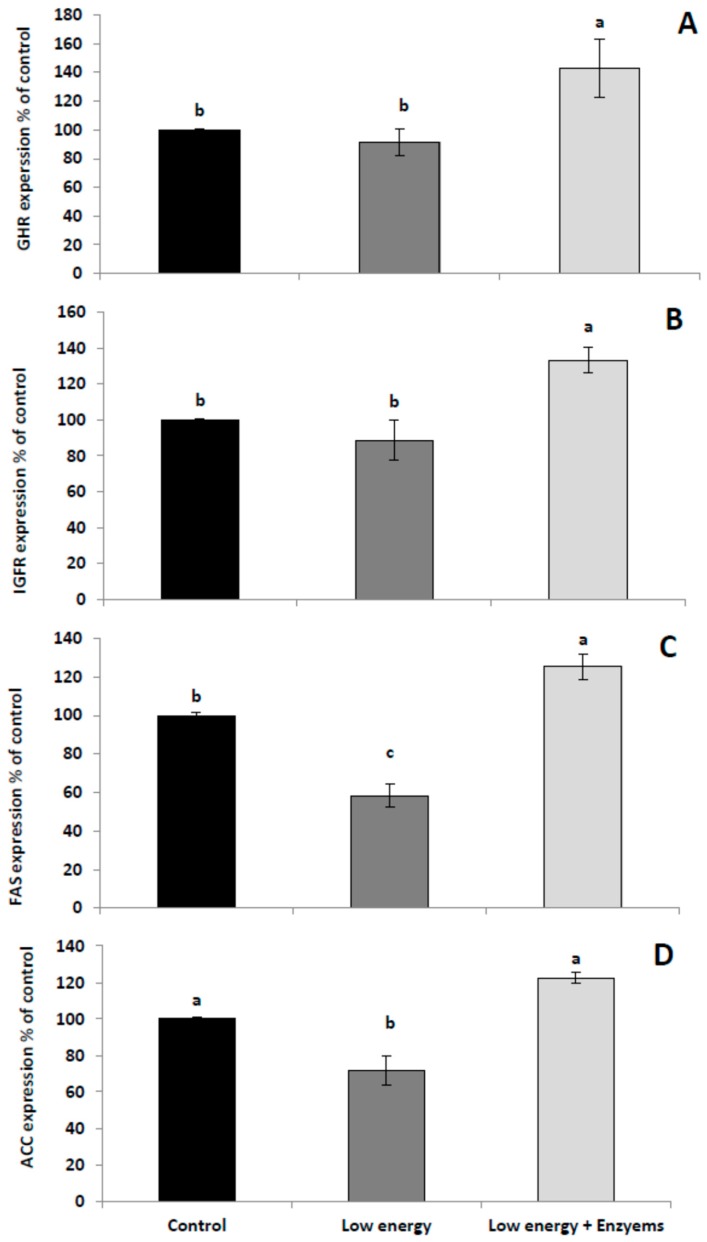
Effect of NSP enzyme supplementation on the gene expression of the growth hormone receptor (GHR) (**A**), insulin-like growth factor receptor (IGFR) (**B**), fatty acid synthesis (FAS) (**C**), and acetyl-coA carboxylase (ACC) (**D**) in broilers. ^a,b^ Mean values with different letters in the same column differ significantly at *p* < 0.05. Values are expressed as means ± standard error. The NSP enzyme used in this experiment was Rovabio^®^ Advance.

**Table 1 animals-09-00467-t001:** Composition of the experimental starter, grower, and finisher diets.

Ingredient	Starter (1–10 days)		Grower (11–25 days)		Finisher (26–35 days)	
	Control	Low Energy	Control	Low Energy	Control	Low Energy
Yellow corn	507	543	548	584	578	613
Soybean meal, 46%	370	352	317	300	280	266
Corn gluten meal, 60%	38	39	50	50	50	47
Soya oil	37	17	41	22	51	32
Calcium carbonate	14.0	14.8	13.8	14.0	12.6	13.4
Dicalcium phosphate	20.0	20.0	17.5	17.5	16.0	16.0
Salt	2.3	2.3	2.4	2.4	2.3	2.3
Sodium sulfate	1.8	1.8	1.6	1.6	1.6	1.6
Dl Methionine, 99%	2.7	2.4	2.0	1.8	1.9	1.8
l-Lysine HCl, 98%	2.5	2.4	2.3	2.3	2.2	2.2
l-Threonine	1.1	1.0	0.7	0.7	0.6	0.6
Choline chloride, 60%	0.8	0.8	0.8	0.8	0.8	0.8
Premix *	2	2	2	2	2	2
Anticoccidia	0.2	0.2	0.2	0.2	0.2	0.2
Anticlostridia	0.1	0.1	0.1	0.1	0.1	0.1
Antimycotoxin biology	0.25	0.25	0.25	0.25	0.25	0.25
Silica	1	1	1	1	1	1
**Chemical Analysis on DM basis**						
AME kcal	3000	2910	3100	3010	3200	3110
Crude protein, %	23.0	22.4	21.5	20.9	20.0	19.4
Fat, %	6.3	4.5	6.9	5.1	7.9	6.2
Digestible LYS, %	1.28	1.24	1.15	1.11	1.06	1.02
Digestible M and C, %	0.95	0.92	0.87	0.84	0.83	0.80
Digestible THR, %	0.86	0.83	0.77	0.74	0.71	0.68
Digestible ARG, %	1.37	1.33	1.25	1.20	1.14	1.11
Digestible ILE, %	0.90	0.87	0.85	0.82	0.77	0.74
Digestible LEU, %	1.87	1.83	1.84	1.78	1.74	1.68
Digestible VAL, %	0.96	0.93	0.91	0.87	0.84	0.81
Calcium, %	0.96	0.96	0.87	0.84	0.81	0.76
Available P, %	0.48	0.48	0.44	0.42	0.41	0.38
Sodium, %	0.16	0.16	0.16	0.16	0.16	0.16
Chloride, %	0.23	0.23	0.23	0.23	0.23	0.23

* Hero mix^®^ (Hero pharm, Cairo, Egypt). Composition (per 3 kg): Vitamin A 12,000,000 IU, vitamin D3 2,500,000 IU, vitamin E 10,000 mg, vitamin K3 2000 mg, vitamin B1 1000 mg, vitamin B2 5000 mg, vitamin B6 1500 mg, vitamin B12 10 mg, niacin 30,000 mg, biotin 50 mg, folic acid 1000 mg, pantothenic acid 10,000 mg, manganese 60,000 mg, zinc 50,000 mg, iron 30,000 mg, copper 4000 mg, iodine 300 mg, selenium 100 mg, and cobalt 100 mg. Diets ingredients and final feed diets were analyzed by chemical analysis in the Adisseo company lab, Antony, France.

**Table 2 animals-09-00467-t002:** Effect of non-starch polysaccharide (NSP) enzyme supplementation on growth performance in broilers.

	Control	Low Energy	Low Energy and Enzymes	*p*-Value
Initial body weight, g	40.9 ± 0.5	40.6 ± 0.7	41.0 ± 0.5	0.29
Final body weight, g/35 d	2382 ± 52 ^a^	2296 ± 57 ^b^	2358 ± 56 ^ab^	0.03
Body weight gain, g/35 d	2341 ± 21.9 ^a^	2255 ± 27.2 ^b^	2317.5 ± 16.4 ^ab^	0.04
Feed intake, g/35 d	3485 ± 34.5	3500 ± 38.9	3410 ± 32.7	0.18
FCR, g/g	1.49 ± 0.02 ^b^	1.55 ± 0.01 ^a^	1.47 ± 0.02 ^b^	0.02
Crude protein digestibility, %	72 ± 3.6 ^a^	64 ± 5.4 ^b^	71 ± 4.8 ^a^	0.05
Crude fat digestibility, %	44 ± 2.4 ^b^	41 ± 3.5 ^b^	52 ± 4.3 ^a^	0.05

^a,b^ Mean values with different letters in the same raw differ significantly at *p* < 0.05. Values are expressed as means ± standard error. The NSP enzyme used in this experiment was Rovabio^®^ Advance. FCR: Feed conversion ratio.

**Table 3 animals-09-00467-t003:** Effect of NSP enzyme supplementation on organ weights (g/100 g BW) in broilers.

	Control	Low Energy	Low Energy and Enzymes	*p*-Value
Carcass	64 ± 0.63	65 ± 0.90	63 ± 0.63	0.33
Breast muscle weight	23 ± 0.49 ^ab^	22 ± 0.63 ^b^	25 ± 1.11 ^a^	0.07
Thigh muscle weight	18 ± 0.37	18 ± 0.34	17 ± 0.29	0.75
Liver weight	1.85 ± 0.07	1.72 ± 0.08	1.82 ± 0.17	0.70
Gizzard weight	1.28 ± 0.08	1.28 ± 0.08	1.19 ± 0.06	0.61
Heart weight	0.37 ± 0.02	0.39 ± 0.01	0.43 ± 0.03	0.27
Spleen weight	0.14 ± 0.02	0.11 ± 0.08	0.12 ± 0.02	0.46
Abdominal fat weight	1.18 ± 0.12 ^a^	0.81 ± 0.09 ^b^	0.89 ± 0.02 ^b^	0.02
Bursa of Fabricius weight	0.13 ± 0.02 ^b^	0.14 ± 0.02 ^b^	0.20 ± 0.02 ^a^	0.04

^a,b^ Mean values with different letters in the same raw differ significantly at *p* < 0.05. Values are expressed as means ± standard error. The NSP enzyme used in this experiment was Rovabio^®^ Advance. BW: Body weight.

**Table 4 animals-09-00467-t004:** Effect of NSP enzyme supplementation on plasma parameters in broilers.

	Control	Low Energy	Low Energy and Enzymes	*p*-Value
GPT, I/U	19 ± 2.33	18 ± 2.61	15 ± 1.05	0.38
GOT, I/U	399 ± 18.10	361 ± 24.70	374 ± 40.60	0.64
Total protein, mg/dL	2.83 ± 0.10	2.98 ± 0.09	3.21 ± 0.12	0.06
Albumin, mg/dL	1.58 ± 0.05	1.59 ± 0.09	1.56 ± 0.08	0.97
Globulin, mg/dL	1.26 ± 0.09 ^b^	1.43 ± 0.04 ^b^	1.7 ± 0.11 ^a^	0.01
Total cholesterol, mg/dL	150 ± 5.3 ^a^	135 ± 3.4 ^b^	133 ± 2.3 ^b^	0.02
Triglycerides, mg/dL	21 ± 2.18	25 ± 2.47	18 ± 3.02	0.20
HDL-cholesterol, mg/dL	56.98 ± 3.01 ^b^	67.47 ± 3.82 ^ab^	71.83 ± 4.44 ^a^	0.04
LDL-cholesterol, mg/dL	79.28 ± 4.2	87.10 ± 11.9	75.81 ± 5.5	0.60
Glucose, mg/dL	193 ± 3.7	193 ± 4.8	199 ± 4.4	0.49
Creatinine, mg/dL	0.47 ± 0.04	0.50 ± 0.02	0.51 ± 0.02	0.49
ND, titer	2.75 ± 0.47 ^b^	3.75 ± 0.47 ^ab^	5.25 ± 0.62 ^a^	0.03
H9N1, titer	0.25 ± 0.03	1.0 ± 0.04	1.25 ± 0.03	0.11

^a,b^ Mean values with different letters in the same raw differ significantly at *p* < 0.05. Values are expressed as means ± standard error. Glutamic oxaloacetic transaminase (GOT); glutamate pyruvate transaminase (GPT); high-density lipoprotein (HDL); low-density lipoprotein (LDL). International Units (I/U). The NSP enzyme used in this experiment was Rovabio^®^ Advance.

**Table 5 animals-09-00467-t005:** Effect of NSP enzyme supplementation on lipid peroxidation, fatty acids, and amino acids in breast muscle of broilers.

	Control	Low Energy	Low Energy and Enzymes	*p*-Value
TBARS, nanomole/g	7.08 ± 0.71 ^a^	5.03 ± 0.46 ^b^	5.35 ± 0.34 ^b^	0.05
Lysine, g/100 g protein	5.89 ± 0.43	5.91 ± 0.49	5.93 ± 0.50	0.99
Methionine, g/100 g protein	1.14 ± 0.04	1.20 ± 0.08	1.20 ± 0.01	0.56
Total lipids, g/100 g muscle	3.13 ± 0.07 ^b^	3.66 ± 0.48 ^ab^	4.98 ± 0.54 ^a^	0.03
Oleic acid, mg/100 g fat	0.124 ± 0.008	0.152 ± 0.012	0.155 ± 0.005	0.96
Linoleic acid, mg/100 g fat	0.226 ± 0.009	0.777 ± 0.065	0.424 ± 0.025	0.63
Linolenic acid, mg/100 g fat	0.095 ± 0.003	0.083 ± 0.008	0.157 ± 0.007	0.70

^a,b^ Mean values with different letters in the same raw differ significantly at *p* < 0.05. Values are expressed as means ± standard error. Thiobarbituric acid reactive substance (TBARS). The NSP enzyme used in this experiment was Rovabio^®^ Advance.

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
