# Peer review of "Effects of Dietary Xylanase and Arabinofuranosidase Combination on the Growth Performance, Lipid Peroxidation, Blood Constituents, and Immune Response of Broilers Fed Low-Energy Diets"

_animals, 2019, doi:10.3390/ani9070467_

Round 1

Reviewer 1 Report

This paper deals with the effect of feeding xylanase arabinofuranosidase on growth performance of broilers fed low energy diets. The author provide information relatively new and of interest to the readers. However several points need be addressed before considering the paper for publication:

General comments

- Clearly state what you considered the experimental unit in the growth performance trial and the digestibility trial.

- It is not clear what is the sense of using a Control group as you have defined it. Indeed you merely use the Control group in the description of results. You do not use it in the discussion. Please delete unless you justify the use.

-Avoid using the terms "numerically enhanced" or similar. Only describe significant results.

-Clearly distinguish a tendency from a significant result in the description of results. You have defined what you were considering as significant, so keep to it.

-When describing results please finish establishing the comparisons " higher than low energy or control". Otherwise the result section is misleading.

- Avoid repeating results in the discussion section. Otherwise write a combined results and discussion section.

- I suggest calculating the effect of feeding enzymes on AME

Specific comments

Simple summary

-Supplementation of enzymes did not cause a higher antibody titer against H9N1. Please delete.

Abstract

-Supplementation of enzymes did not cause a higher did not cause a higher body weight gain (SEE TABLE 2). Please delete.

-You are not measuring fat or protein utilization. You are measuring digestibility. Please reword.

-Supplementation of enzymes did not cause a higher antibody titer against H9N1. Please delete.

Materials and methods

2.1. animals and experimental design

-Provide information on the pen`s dimensions or the stocking density 

-Did you record mortality?

-Provide a wider descrip of the composition of robavio

- Provide clear information on how the enzymes were formulated in the concentrate

- Specify the duration of each growth phase (starter, grower, finisher)

- Provide information on the format of the concentrate: crumble, pellet...in each growth phase. Was it provided ad libitum?

-Did you measure body weight on individual or group basis?

-How did you measure feed intake? on group basis?

- You have 8 replicates. Were they group fed? if so, please clearlty state.

-If you have 8 replicates and they were group fed (?), then your experimental unit is the pen, and consequently you cannot consider 12 birds/treatment in your statistical analysis. If you did, data needs to be statistically analyzed again.

-Were all the 12 chicks sampled from the same pen? If not, from how many pens per treatment?

- Table 1. Indicate if ingredient formulation and chemical analyses are given as fed or on DM basis. Indicate which variables you measured and which you calculated. For instance did you measure digestible lys? or did you use a calculation?

2.2. nutrient digestibility

- Were broilers housed individually?

-What did you consider as experimental unit?

-Provide information on the cage's dimensions

-Provide information on the experimental design: adaptation period? sampling period?

-You did not measure nitrogen or fat utilization/retention. You are measuring digestibility.

- You measured fat and cp digestibility. It is surprising that you did not measure DM, OM or fiber digestibility. Any reasons?

2.5. RNA analysis

- In which sample did you perform RNA analysis? because you sampled breast and thigh

2.6. statistical analysis

-"...treatment conditions were significantly different...p<0.05". Not true. You are considering significant differences at P=0.073 or P=0.063. You clearly stated what you were considering. Either keep to it or do not state it.

3.Results

-It would be interesting to provide growth results per productive phase: starter, grower, finisher, whole cycle.

-Table 2 does not show that feeding enzymes  increased body weight gain (low energy (b), low energy+enzymes (ab). Differences between control (a) and low energy (b).

-Table 2. Define FCR. Use international system units here and elsewhere: g instead of grams. Reword CP and fat utilization as suggested before. Reduce decimal in initial body weight.

-Table 3. Define bw. Reduce decimal use in breast and thigh muscle weight

-Table 4 does not show a significant effect of enzymes on total globulin, HDL-cholesterol or ND titer compared to low energy. Please reword

-Table 5 does not show a significant effect of enzymes on TBARS or total lipids compared with low energy. Please reword

-Figure 1. Try to use the same pattern of comparison . In the previous description of results you describe the effect of using enzymes, but for the mRNA expression you describe the effect of low energy diets without enzymes.

Discussion

-Discussion needs to be entirely reworded. 

Author Response

Dear Prof. Editor-in-Chief, Animals

cc,

Ms. Zarol Han, Managing Editor

Regarding to the manuscript entitled "Effects of Dietary Xylanase and Arabinofuranosidase Combination on the Growth Performance, Lipid Peroxidation, Blood Constituents, and Immune Response of Broilers Fed Low-Energy Diets"   

Thank you in advance for your time and effort on reviewing our work.

A list of modifications according to the suggestions and comments of the reviewers is attached below. We are fully appreciated the valuable suggestions of the reviewers. Moreover, we are proud that our study has good discussion by the reviewers.

Sincerely Yours,

Ahmed Ali Mahmoud Saleh, PhD
Professor
Department of Poultry Production
Faculty of Agriculture,
Kafrelsheikh University, Egypt.

List of modifications according to the suggestions and comments of reviewers:

(Revisions related to reviewers’ comments are shown in red in the revised manuscript)

The authors appreciate the comments from the reviewers. The manuscript has been revised in accordance with their requests. We do our best to take all comments in consideration, incorporating them into the revised manuscript as indicated in our responses to the reviewers.

Comments and Suggestions for Authors

This paper deals with the effect of feeding xylanase arabinofuranosidase on growth performance of broilers fed low energy diets. The author provide information relatively new and of interest to the readers. However several points need be addressed before considering the paper for publication:

General comments

- Clearly state what you considered the experimental unit in the growth performance trial and the digestibility trial.

Response: Thank you for your suggestion

The experimental unit for body weight and digestibility experiment was the individual, while; the experimental unit for feed intake was the pen.

In P3 L 109-110, We stated that "Body weight was measured individually every week, and feed intake was measured daily as group basis per pen throughout the experimental period."

In P3 L 110-112, We stated that "At 32 days, all birds were weighed individually and sorted from the smallest to the heavy weight, then 12 birds/treatment in the same middle weights from the group weights were transferred to special batteries owing individually cages to make digestibility experiment."

- It is not clear what is the sense of using a Control group as you have defined it. Indeed you merely use the Control group in the description of results. You do not use it in the discussion. Please delete unless you justify the use.

We used Control group in the description of results in a lot of positions (Lines 165, 175, 185, 200 & 210) to explain the importance of the obtained results.

-Avoid using the terms "numerically enhanced" or similar. Only describe significant results.

We avoided using terms "numerically enhanced" or "similar".

-Clearly distinguish a tendency from a significant result in the description of results. You have defined what you were considering as significant, so keep to it.

We clearly stated the importance of the significant results due to Xyl and Abf enzymes supplementation. There are significant improvements in body weight gain (L 166), FCR (L 166), crude protein digestibility (L 166), crude fat digestibility (L 166), breast muscle (L 173) and bursa of Fabricius (L 173) due to Xyl and Abf enzymes supplementation, which indicated a positive effect.

-When describing results please finish establishing the comparisons " higher than low energy or control". Otherwise the result section is misleading.

In Lines 165, 175, 185, 189, 201& 210: We compared the obtained results with the control group or with low energy diets.

- Avoid repeating results in the discussion section. Otherwise write a combined results and discussion section.

In Discussion, we avoided repeating the results and we concentrated in the main obtained results and we offer interpretations for the obtained results.

Specific comments

Simple summary

-Supplementation of enzymes did not cause a higher antibody titer against H9N1. Please delete.

Response: Thank you for your suggestion

We deleted it P1 L28.

Abstract

-Supplementation of enzymes did not cause a higher did not cause a higher body weight gain (SEE TABLE 2). Please delete.

Response: Thank you for your suggestion

We corrected it P1 L37."Feed intake was decreased by low energy diet, leading to an enhancement in feed efficiency enzyme supplementation in low energy diet (p < 0.015)."

-You are not measuring fat or protein utilization. You are measuring digestibility. Please reword.

Response: Thank you for your suggestion

We reword it P1 L38. (Both protein and fat digestibilities were improved due to enzyme supplementation.)

-Supplementation of enzymes did not cause a higher antibody titer against H9N1. Please delete.

Response: Thank you for your suggestion

We deleted it. In P1 L40-41.

Materials and methods

2.1. animals and experimental design

-Provide information on the pen`s dimensions or the stocking density 

Response: Thank you for your suggestion

We added it in P2 L91 ((stocking density was 10 birds/m²)

-Did you record mortality?

Yes, we record mortality and we have only three dead birds one each treatment at the first three days.

-Provide a wider descrip of the composition of robavio

Response: Thank you for your suggestion

We added it in P3 L100-102 (This enzyme was industrially created by fermentation of Talaromyces versatilis (IMI378536 and DSM26702; Adisseo France S.A.S. proprietary strains) and the main enzyme activity in Enz comes from Xyl and Abf)

- Provide clear information on how the enzymes were formulated in the concentrate

Response: Thank you for your suggestion

We added this information in P3L102 (The enzyme was mixed in the premix which used in the third treatment formula.)

- Specify the duration of each growth phase (starter, grower, finisher)

Response: Thank you for your suggestion

We added it in Table 2 "Starter (1-10 days), Grower (11-25 days), and Finisher (26-35 days)"

- Provide information on the format of the concentrate: crumble, pellet...in each growth phase. Was it provided ad libitum?

Response: Thank you for your suggestion

We added it P3 L103-104 (The diets were proved to birds ad libitum; starter diets were crumble form however, grower and finisher diets were pellet form)

-Did you measure body weight on individual or group basis?

Response: Thank you for your suggestion

We answered P3 L107 (Body weight was measured individually every week,)

-How did you measure feed intake? on group basis?

Response: Thank you for your suggestion

We added answered it P3 L 108 (feed intake was measured daily as group basis per pen throughout the experimental period.)

- You have 8 replicates. Were they group fed? if so, please clearlty state.

Response: Thank you for your suggestion

We added answered it P3 L 108 (feed intake was measured daily as group basis per pen throughout the experimental period.)

-If you have 8 replicates and they were group fed (?), then your experimental unit is the pen, and consequently you cannot consider 12 birds/treatment in your statistical analysis. If you did, data needs to be statistically analyzed again.

Response: Thank you for your suggestion

We described this point and how can we choose the birds P3 L110-112. (At 32 days, all birds were weighed individually and sorted from the smallest to the heavy weight, then 12 birds/treatment in the same middle weights from the group weights were transferred to special batteries owing individually metabolic cages to make digestibility experiment then these birds slaughtered, and dissected to gauge the weights of the breast muscle, thigh muscle, liver, gizzard, heart, spleen, abdominal fat, and bursa of Fabricius)

-Were all the 12 chicks sampled from the same pen? If not, from how many pens per treatment?

Response: Thank you for your suggestion

We described this point up and how can we choose the birds P3 L108-112 the birds were chosen from different bens in each treatment.

- Table 1. Indicate if ingredient formulation and chemical analyses are given as fed or on DM basis. Indicate which variables you measured and which you calculated. For instance did you measure digestible lys? or did you use a calculation?

Response: Thank you for your suggestion

We added more information in Table1, we illustrated that "Chemical analysis on DM basis". Also, in footnote, we mentioned that (Diets ingredients and final feed diets were analyzed by chemical analysis in Adisseo company lab, France.) Also, all digestible amino acids were determined in the ingredients and in the final diets after carrying out the chemical analysis in the faces and ileum contents.

2.2. nutrient digestibility

- Were broilers housed individually?

Response: Thank you for your suggestion

Yes, in digestibility experiment, broilers were housed individually. In P4 L 127-129 " In the last three days of the experiment, excreta were gathered and weighted from 12 birds per treatment, where broilers were housed individually in special metabolic cages (40×40×50 cm) for digestibility tests."

Also, we mentioned this in P3 L110-114.

-What did you consider as experimental unit?

Response: Thank you for your suggestion

The experimental unit for body weight, carcass measurements and digestibility experiment was the individual, while, the experimental unit for feed intake was the pen.

In P3 L 109-110, We stated that "Body weight was measured individually every week, and feed intake was measured daily as group basis per pen throughout the experimental period."

In P3 L 110-112, We stated that "At 32 days, all birds were weighed individually and sorted from the smallest to the heavy weight, then 12 birds/treatment in the same middle weights from the group weights were transferred to special batteries owing individually cages to make digestibility experiment."

-Provide information on the cage's dimensions

Response: Thank you for your suggestion

We added more information P4 L128 "(40×40×50 cm) ".

-Provide information on the experimental design: adaptation period? sampling period?

Response: Thank you for your suggestion

We added more information P4 L129-131 (During these three days the birds and feed intake were weighted daily, and faces extracted were collected and weighted and stored in freezer. After digestibility experiment period all samples were dried in a drying oven at 60 °C for 24 h)

-You did not measure nitrogen or fat utilization/retention. You are measuring digestibility.

Response: Thank you for your suggestion

We corrected it P4 L134.

- You measured fat and cp digestibility. It is surprising that you did not measure DM, OM or fiber digestibility. Any  reasons?

Response: Thank you for your suggestion

We analyses feed and feces for DM, CP , EE, FCU etc… but these results will publish in another paper in this paper we only add protein and fat because the objective in this paper depend on the matrix value in the enzyme affect on CP and ME.

2.5. RNA analysis

- In which sample did you perform RNA analysis? because you sampled breast and thigh

Response: Thank you for your suggestion

We added it P4 L151 ( Each breast muscle sample)

2.6. statistical analysis

-"...treatment conditions were significantly different...p<0.05". Not true. You are considering significant differences at P=0.073 or P=0.063. You clearly stated what you were considering. Either keep to it or do not state it.

Response: Thank you for your suggestion

We corrected it P4L162.

3.Results

-It would be interesting to provide growth results per productive phase: starter, grower, finisher, whole cycle.

Response: Thank you for your suggestion

We think this will be deep results and we will add in the next paper.

-Table 2 does not show that feeding enzymes  increased body weight gain (low energy (b), low energy+enzymes (ab). Differences between control (a) and low energy (b).

Response: Thank you for your suggestion

We corrected it P5L164-167. (Table 2 show that feeding low-energy diets decreased body weight gain significantly compared to the control group, while, dietary supplementation with Xyl and Abf enzymes increased body weight gain and improved FCR, crude protein digestibility, and crude fat digestibility significantly (p < 0.05).

-Table 2. Define FCR. Use international system units here and elsewhere: g instead of grams. Reword CP and fat utilization as suggested before. Reduce decimal in initial body weight.

Response: Thank you for your suggestion

We corrected all these inquires in Table 2 and in footnote.

-Table 3. Define bw. Reduce decimal use in breast and thigh muscle weight

Response: Thank you for your suggestion

We corrected all these inquires in Table 3.

-Table 4 does not show a significant effect of enzymes on total globulin, HDL-cholesterol or ND titer compared to low energy. Please reword

Response: Thank you for your suggestion

We corrected it P5 L 184.

-Table 5 does not show a significant effect of enzymes on TBARS or total lipids compared with low energy. Please reword

Response: Thank you for your suggestion

We corrected it P6 L 201.

-Figure 1. Try to use the same pattern of comparison . In the previous description of results you describe the effect of using enzymes, but for the mRNA expression you describe the effect of low energy diets without enzymes.

Response: Thank you for your suggestion

We corrected it P7 L 209- 214. "The mRNA expressions of growth hormone receptor (GHR), insulin-like growth factor receptor (IGFR), and fatty acid synthesis (FAS) were significantly increased by adding Xyl and Abf enzymes to low-energy diets in comparison with the control group (Figure 1A–C). Meanwhile, acetyl-coA carboxylase (ACC) was increased by Xyl and Abf enzyme supplementation to low energy diet but these differences was not significant in comparison with the control group (Figure 1D)."

Discussion

-Discussion needs to be entirely reworded. 

Response: Thank you for your suggestion

We reworded it and we concentrated in the main obtained results and we offer interpretations for the obtained results.

Comments and Suggestions for Authors

Abstract:

The growth – please change as follow: the performance

Response: Thank you for your suggestion

We corrected it (P1 L31).

Digestibility of what?

Response: Thank you for your suggestion

We added it P1 L31 (protein and fat digestibilities)

Not blood constituents – please change as follow: the plasma biochemical traits

Response: Thank you for your suggestion

We corrected it P1 L32 " the plasma biochemical traits ".

If it is possible the results mentioned in the abstract should have a p-value

Response: Thank you for your suggestion

We added p-values (P1 L 38, 39& 42).

What does it mean protein and fat utilization? Is it nitrogen retention and total tract digestibility of ether extract? It should be clear.

Response: Thank you for your suggestion

We corrected it to digestibility (P1 L 31, 38 & 45).

Comment 1: In the Abstract section, more information about material and methods should appear, for instance: birds number,  hybrid, sex, replication number, etc.

Response: Thank you for your suggestion

We added it  in P1 L32: " A total of 480, un-sexed one-day-old broilers (Ross 308) were randomly divided into three treatments with 8 replicates"

Introduction:

Please explain - amendment of gut microbiota; The authors meant the positive modulations of microecosystem of gastrointestinal tract microbiota?

Response: Thank you for your suggestion

In P2 L 68: We complete the senate’s as following (by improved the beneficial microbes)

In the aim: the total tract ether extract digestibility and retention of nitrogen, as well as biochemical plasma traits

Response: Thank you for your suggestion

We corrected it in P2 L88-89 (fat and protein digestibilities,……. biochemical plasma traits,)

Material and Methods

The Authors should add information about ethical committee permission.

Response: Thank you for your suggestion

We added it P3L91-92 (The study was approved by the Ethics Committee of Local Experimental Animals Care Committee and conducted in accordance with the guidelines of Kaferelsheikh University, Egypt)

The Authors omitted information about the sex of birds. What was the sex? Male or female? If both what was the sex ration in each replication?

Response: Thank you for your suggestion

We used un-sexed birds one day old as commercial chickens (P3 L94).

Did birds are feed ad libitum?

Response: Thank you for your suggestion

Yes , we mentioned this in P3L106 " ad libitum ".

What was the size of the bins?

Response: Thank you for your suggestion

We added these information P3 L94-95 (stocking density was 10 birds/m²)

Where did blood samples were collected from? Wing vein? The collection was made immediately before slaughtering? Birds were fed before or the Authors made some starving period?

Response: Thank you for your suggestion

We added more information about blood sample collecting, however, we did not made a starving period because we need the birds will eating until last second for collecting more feces for digestibility. Also, we think the starving will make some stress for the birds which will change in blood glucose and lipids. In P3 L116-118. (Blood samples were collected from wing vein immediately before slaughtering and gathered into heparinized test tubes and rapidly centrifuged)

Comment 2: Why the Authors used only 12 bird from each group where the number of replicates was 8? What about 4 additional birds, how these were chosen? From the Reviewer point of view, it should be taken only 8 birds (1 chicken from each box) or 16 (2 birds from each replication).

Response: Thank you for your suggestion

We agree for your point however, we added more information for this point in P3 L110-114. "At 32 days, all birds were weighed individually and sorted from the smallest to the heavy weight, then 12 birds/treatment in the same middle weights from the group weights were transferred to special batteries owing individually cages to make digestibility experiment then these birds were slaughtered, and dissected to gauge the weights of the breast muscle, thigh muscle, liver, gizzard, heart, spleen, abdominal fat, and bursa of Fabricius."

Please, add information about the measuring of internal organs. It was in relation to body weight?

Response: Thank you for your suggestion

We added more information P 3L115-116 (All organs were weighted and described as ratio of the body weight)

Comment 3: The Reviewer understand that the product which was examined Is probably patented by Adisseo company, however, it is hard to check, reviewed the results without any information about the product. In the case of exogenous enzymes used as feed additives, it is crucial to know, e.g., the activities of each enzyme preparation, as well as the microorganism produces.

Response: Thank you for your suggestion

We added more information P 3 L102-105 (This enzyme was industrially created by fermentation of Talaromyces versatilis (IMI378536 and DSM26702; Adisseo France S.A.S. proprietary strains) and the main enzyme activity in Enz comes from Xyl and Abf)

What do the Authors mean using – anticoccidia, aciticlostridia, and antimycotoxin biology, in the diet composition? Please, expand.

Response: Thank you for your suggestion

We believe that this experiment will be applied in the commercial field. Moreover, in all commercial feed in Egypt they used all these additives.

AME kcal mentioned in the diets is only calculated, what about analyzed value? It is important in terms of the effect of enzymes.

Response: Thank you for your suggestion

All the ingredients were analyzed chemically then, we make the diets formulas according these analyses. Also, final feed diets analyzed chemically to be sure the calculated numbers same of chemicals analysis results.

Nutrient digestibility

The Authors mentioned before that the birds were kept in bins. In this paragraph, the Authors mentioned that 10 birds per treatment were kept in special batteries. These birds were the same as before of other chickens which was used for digestibility trial? Please, explain in detail.

Response: Thank you for your suggestion

We added more information about the digestibility experiment P4 L126-130. However, we used 12 birds for the digestibility and we corrected the number (In the last three days of the experiment, excreta were gathered and weighted from 12 birds per treatment, where broilers were housed individually in special metabolic cages (40×40×50 cm) for digestibility tests. During these three days, the birds and feed intake were weighted daily, and faces extracted were collected and weighted and stored in freezer. After digestibility experiment period all samples were dried in a drying oven at 60 °C for 24 h.)

What does it mean ‘special batteries’? Metabolic cages? Please, add the size and other conditions.

Response: Thank you for your suggestion

We added this information in P4 L129-130 "where broilers were housed individually in special metabolic cages (40×40×50 cm) for digestibility tests."

Please, add AOAC numbers (Kjeldahl and Soxhlet).

Response: Thank you for your suggestion

We added this information P4L135 "(AOAC 945.38F and 920.39C, respectively)."

In the manuscript should be emphasized that the Authors measured total tract digestibility of ether extract, as well as retention of nitrogen. Please, double check the whole paper.

Response: Thank you for your suggestion

We corrected it throughout all the manuscript.

Results

Table 2: It is not true that the 3rd group increased BWG significantly. It is at the same level a low energy diet without enzymes. Please, correct!

Response: Thank you for your suggestion

We corrected  P5 L 165-168 (The data presented in Table 2 show that feeding low-energy diets decreased body weight gain significantly compared to the control group, while, dietary supplementation with Xyl and Abf enzymes increased body weight gain and improved FCR, crude protein digestibility, and crude fat digestibility significantly (p < 0.05).

The table should be informed which enzymes were used in the study. More information should be added. The table has to inform about the design of the experiment.  Tables need to be an independent figure and should be clear without text from the manuscript. It is a comment for each table.

Response: Thank you for your suggestion

We added the name of the enzyme used under all tables and figure "The NSP enzymes used in this experiment was Rovabio® Advance"

Please, change the ‘crude protein utilizations’ as well as ‘crude fat utilization’ as the Reviewer mentioned above.

Response: Thank you for your suggestion

We corrected it in the table and in the text.

Table 3:

There is no significant change in breast muscle weight; p = 0.073! Please, correct.

Response: Thank you for your suggestion

We corrected it (P5 L174).

The unit, i.e., g/100g BW could be mentioned in the title, and exclude from the table.

Response: Thank you for your suggestion

We added in the table title and exclude from the table.

Table 4:

The total protein is not significantly increased, there is no significance (p=0.063!). Please,

 correct.

Response: Thank you for your suggestion

We corrected it

The sentence: ‘Plasma total protein, globulin, and HDL-cholesterol were significantly increased.’ Is no informative, please remodeled.

Response: Thank you for your suggestion

We corrected it (P6 L184).

The sentence: ‘However, plasma total cholesterol was significantly reduced by the addition of Xyl and Abf enzymes to low-energy diets’ is not true because the low energy diet caused the same effect.

Response: Thank you for your suggestion

We corrected it as we compared it with the control group (P6 L187)

Comment 4: Did the birds are vaccinated before the experiment?

Response: Thank you for your suggestion

Yes, the birds have one day vaccination.

Table 5:

The title should be remodeled with a native speaker.

Response: Thank you for your suggestion

We corrected it.

Comment 5: the sentence: ‘The mRNA expressions of growth hormone receptor (GHR), insulin-like growth factor receptor (IGFR), fatty acid synthesis (FAS), and acetyl-coA carboxylase (ACC) were significantly decreased in low-energy diets without Xyl and Abf enzyme supplementation.’ Is not true… please notice that the low energy diet did not differ in comparison to control group in  terms of GHR, IGFR… It should be rewritten.

Response: Thank you for your suggestion

We rewritten it P7 L210-215 " The mRNA expressions of growth hormone receptor (GHR), insulin-like growth factor receptor (IGFR), and fatty acid synthesis (FAS) were significantly increased by adding Xyl and Abf enzymes to low-energy diets in comparison with the control group (Figure 1A–C). Meanwhile, acetyl-coA carboxylase (ACC) was increased by Xyl and Abf enzyme supplementation to low energy diet but these differences was not significant in comparison with the control group (Figure 1D). "

Reviewer 2 Report

Abstract:

The growth – please change as follow: the performance

Digestibility of what?

Not blood constituents – please change as follow: the plasma biochemical traits

If it is possible the results mentioned in the abstract should have a p-value

What does it mean protein and fat utilization? Is it nitrogen retention and total tract digestibility of ether extract? It should be clear.

Comment 1: In the Abstract section, more information about material and methods should appear, for instance: birds number,  hybrid, sex, replication number, etc.

Introduction:

Please explain - amendment of gut microbiota; The authors meant the positive modulations of microecosystem of gastrointestinal tract microbiota?

In the aim: the total tract ether extract digestibility and retention of nitrogen, as well as biochemical plasma traits

Material and Methods

The Authors should add information about ethical committee permission.

The Authors omitted information about the sex of birds. What was the sex? Male or female? If both what was the sex ration in each replication?

Did birds are feed ad libitum?

What was the size of the bins?

Where did blood samples were collected from? Wing vein? The collection was made immediately before slaughtering? Birds were fed before or the Authors made some starving period?

Comment 2: Why the Authors used only 12 bird from each group where the number of replicates was 8? What about 4 additional birds, how these were chosen? From the Reviewer point of view, it should be taken only 8 birds (1 chicken from each box) or 16 (2 birds from each replication).

Please, add information about the measuring of internal organs. It was in relation to body weight?

Comment 3: The Reviewer understand that the product which was examined Is probably patented by Adisseo company, however, it is hard to check, reviewed the results without any information about the product. In the case of exogenous enzymes used as feed additives, it is crucial to know, e.g., the activities of each enzyme preparation, as well as the microorganism produces.

What do the Authors mean using – anticoccidia, aciticlostridia, and antimycotoxin biology, in the diet composition? Please, expand.

AME kcal mentioned in the diets is only calculated, what about analyzed value? It is important in terms of the effect of enzymes.

From the Reviewer point of view, the diets should be calculated

Nutrient digestibility

The Authors mentioned before that the birds were kept in bins. In this paragraph, the Authors mentioned that 10 birds per treatment were kept in special batteries. These birds were the same as before of other chickens which was used for digestibility trial? Please, explain in detail.

What does it mean ‘special batteries’? Metabolic cages? Please, add the size and other conditions.

Why the Authors used an oven for preparing samples to calculate the retention? Please, add the reference to this method. Usually, the samples for digestibility/retention is freeze-dried (lyophilized).

Please, add AOAC numbers (Kjeldahl and Soxhlet).

In the manuscript should be emphasized that the Authors measured total tract digestibility of ether extract, as well as retention of nitrogen. Please, double check the whole paper.

Results

Table 2: It is not true that the 3rd group increased BWG significantly. It is at the same level a low energy diet without enzymes. Please, correct!

The table should be informed which enzymes were used in the study. More information should be added. The table has to inform about the design of the experiment.  Tables need to be an independent figure and should be clear without text from the manuscript. It is a comment for each table.

Please, change the ‘crude protein utilizations’ as well as ‘crude fat utilization’ as the Reviewer mentioned above.

Table 3:

There is no significant change in breast muscle weight; p = 0.073! Please, correct.

The unit, i.e., g/100g BW could be mentioned in the title, and exclude from the table.

Table 4:

The total protein is not significantly increased, there is no significance (p=0.063!). Please, correct.

The sentence: ‘Plasma total protein, globulin, and HDL-cholesterol were significantly increased.’ Is no informative, please remodeled.

The sentence: ‘However, plasma total cholesterol was significantly reduced by the addition of Xyl and Abf enzymes to low-energy diets’ is not true because the low energy diet caused the same effect.

Comment 4: Did the birds are vaccinated before the experiment?

Table 5:

The title should be remodeled with a native speaker.

Comment 5: the sentence: ‘The mRNA expressions of growth hormone receptor (GHR), insulin-like growth factor receptor (IGFR), fatty acid synthesis (FAS), and acetyl-coA carboxylase (ACC) were significantly decreased in low-energy diets without Xyl and Abf enzyme supplementation.’ Is not true… please notice that the low energy diet did not differ in comparison to control group in  terms of GHR, IGFR… It should be rewritten.

Author Response

(The authors gave the same response as above.)

Round 2

Reviewer 1 Report

Authors have improved the manuscript as requested. The manuscript is clear from a methodological point of view. However, some points need to be addressed before considering it acceptable. I would appreciate very much if authors could provide a totally lined version next time.

English editing:

I suggest a thorough revision by a native English speaker. For instance:

-Line 68, 105-107, 111-114 and elsewhere.

Other comments:

-Line 62: provide the ethical approval code

Table 1. Delete decimals for corn gluten meal, soya oil, use 1 decimal for salt, sodium sulfate, methionine, lysine and threonine. Delete total/kg. Use one decimal for crude protein and fat and two for the other parameters.

-Line 137. It is not nitrogen retention. It is nitrogen digestibility. Here, the same should be indicated for fat digestibility.

Table 2. Round to three decimals for P values. Means within a row differ, not within a column

Table 3. BW instead of bw. Delete decimals in carcass, breast and thigh. Round to two decimals in all others. Round to three decimals for P values. Means within a row differ, not within a column

Table 4. Delete decimals in GOT, cholesterols, triglycerides and glucose. Round to 2 decimals in all others. Round to three decimals for P values. Means within a row differ, not within a column

Table 5. Use IS units: g instead of gram. Add units to oleic, linoleic and linolenic acids. Round to two decimals in tbars, aminoacids and total lipids. Round to 3 decimals in all others. Delete units in the footnote. Means within a row differ, not within a column.

-Avoid calling tables and giving P values in the discussion, otherwise go for a combined results and discussion section ( the discussion is not lined, I cannot be more specific).

-Avoid the term "numerical" for explanations. Still in use in the discussion.

-Results are repeated in the discussion. Try to reword the discussion avoiding repeating results. Otherwise, go for a combined section.

Author Response

Dear Prof. Editor-in-Chief, Animals

cc,

Ms. Zarol Han, Managing Editor

Regarding to the manuscript entitled "Effects of Dietary Xylanase and Arabinofuranosidase Combination on the Growth Performance, Lipid Peroxidation, Blood Constituents, and Immune Response of Broilers Fed Low-Energy Diets"   

Thank you in advance for your time and effort on reviewing our work.

A list of modifications according to the suggestions and comments of the reviewers is attached below. We are fully appreciated the valuable suggestions of the reviewers. Moreover, we are proud that our study has good discussion by the reviewers.

Sincerely Yours,

Ahmed Ali Mahmoud Saleh, PhD
Professor
Department of Poultry Production
Faculty of Agriculture,
Kafrelsheikh University, Egypt.

List of modifications according to the suggestions and comments of reviewers:

(Revisions related to reviewers’ comments are shown in red in the revised manuscript)

The authors appreciate the comments from the reviewers. The manuscript has been revised in accordance with their requests. We do our best to take all comments in consideration, incorporating them into the revised manuscript as indicated in our responses to the reviewers.

Comments and Suggestions for Authors

Authors have improved the manuscript as requested. The manuscript is clear from a methodological point of view. However, some points need to be addressed before considering it acceptable. I would appreciate very much if authors could provide a totally lined version next time.

English editing:

I suggest a thorough revision by a native English speaker. For instance:

-Line 68, 105-107, 111-114 and elsewhere.

Response: Thank you for your suggestion

English language of the manuscript was performed and edited by MDPI. We attached the certificate for English editing.

Moreover, we corrected the sentences L 68, 105-106, 111-114.

Other comments:

-Line 92: provide the ethical approval code

Response: Thank you for your suggestion

We added the ethical approval code P3L94   (Number 4/2016 EC).

Table 1. Delete decimals for corn gluten meal, soya oil, use 1 decimal for salt, sodium sulfate, methionine, lysine and threonine. Delete total/kg. Use one decimal for crude protein and fat and two for the other parameters.

Response: Thank you for your suggestion

We corrected Table.1.

-Line 137. It is not nitrogen retention. It is nitrogen digestibility. Here, the same should be indicated for fat digestibility.

Response: Thank you for your suggestion

We corrected it P4 L137.

Table 2. Round to three decimals for P values. Means within a row differ, not within a column

Response: Thank you for your suggestion

We corrected Table. 2. We corrected to two decimals for P values. Also, we corrected that "letters in the same raw "

Table 3. BW instead of bw. Delete decimals in carcass, breast and thigh. Round to two decimals in all others. Round to three decimals for P values. Means within a row differ, not within a column

Response: Thank you for your suggestion

We corrected Table. 3 including using BW instead of bw. We deleted decimals in carcass, breast and thigh. We used two decimals in the Table. Also, we corrected that "letters in the same raw "

Table 4. Delete decimals in GOT, cholesterols, triglycerides and glucose. Round to 2 decimals in all others. Round to three decimals for P values. Means within a row differ, not within a column

Response: Thank you for your suggestion

We corrected Table. 4. We deleted decimals in GOT, cholesterols, triglycerides and glucose. We used two decimals in the Table. Also, we corrected that "letters in the same raw "

Table 5. Use IS units: g instead of gram. Add units to oleic, linoleic and linolenic acids. Round to two decimals in tbars, aminoacids and total lipids. Round to 3 decimals in all others. Delete units in the footnote. Means within a row differ, not within a column.

Response: Thank you for your suggestion

We corrected Table 5 . We used g instead of gram.  We added to oleic, linoleic and linolenic acids "mg/100 g fat". We deleted units in the footnote.  Also, we corrected that "letters in the same raw "

-Avoid calling tables and giving P values in the discussion, otherwise go for a combined results and discussion section ( the discussion is not lined, I cannot be more specific).

We combined results with discussion in one section to be easy for reading.

-Avoid the term "numerical" for explanations. Still in use in the discussion.

We deleted "numerical" in whole the manuscript.

-Results are repeated in the discussion. Try to reword the discussion avoiding repeating results. Otherwise, go for a combined section.

Response: Thank you for your suggestion

We combined results with discussion in one section to be easy for reading.

Comments and Suggestions for Authors

From the Reviewer point of view, the study design has not carried out correctly. The Authors claimed that the used birds were not sexed, and were randomly allotted to 3 dietary treatments. So it is not possible to set sex ratio 50:50 in each replication. 

It is well-documented that the sex has a crucial role in the growth performance, digestibility of nutrients, and also biochemical blood parameters, in general, physiological response. 

Response: Thank you for your suggestion

We agree with you points that the sex has a crucial role in the growth performance, digestibility of nutrients, and also biochemical blood parameters. But in Egyptian market it is impossible to get one-day-old sexed broilers. In the commercial market, only unsexed broilers was available. However, we were very careful in this point. We used males for digestibility of nutrients, and also biochemical blood parameters.  At 32 day of the experiment we choose the birds in the average weight of all treatments and we used only males for the analysis as we mentioned this in P3L112 and P4 L129.

Moreover, in several recent published papers, they using unsexed broiler chicks in similar parameters as follow:

2019 Poultry Science, pez099, https://doi.org/10.3382/ps/pez099

2018 Poultry Science, Pages 1177–1188, https://doi.org/10.3382/ps/pex404

2017 Poultry Science, Pages 4182–4192, https://doi.org/10.3382/ps/pex173

2017 British Poultry Science, 512–516. https://doi.org/10.1080/00071668.2017.1332405

Reviewer 2 Report

From the Reviewer point of view, the study design has not carried out correctly. The Authors claimed that the used birds were not sexed, and were randomly allotted to 3 dietary treatments. So it is not possible to set sex ratio 50:50 in each replication. 

It is well-documented that the sex has a crucial role in the growth performance, digestibility of nutrients, and also biochemical blood parameters, in general, physiological response. 

Author Response

(The authors gave the same response as above.)

Round 3

Reviewer 2 Report

The Authors explained the problem of the usage of un-sexed birds in the experiment. The Reviewer accepts the analyses which have done only on males birds.